# A Novel $\alpha$-BaTeMo$_2$O$_9$ Acousto-Optic Switch for Generating Stable 639 nm Pulsed Laser

Ke Zhang [1], Feifei Guo [2], Yicheng Jin [1], Kuan Li [1], Lihua Meng [1,*], Peifu Wang [1], Shande Liu [1,*], Zeliang Gao [2,*] and Xutang Tao [2]

1   College of Electronic and Information Engineering, Shandong University of Science and Technology, Qingdao 266590, China
2   State Key Laboratory of Crystal Materials, Shandong University, Jinan 250100, China
*   Correspondence: mlhtai@163.com (L.M.); pepsl_liu@163.com (S.L.); gaozeliang@sdu.edu.cn (Z.G.)

**Abstract:** In this paper, an acousto-optic (AO) Q-switch based on $\alpha$-BaTeMo$_2$O$_9$ ($\alpha$-BTM) crystal is designed and further applied to generate a laser pulse at 639 nm for the first time. The $\alpha$-BTM AO Q-switch demonstrates a large diffraction angle of 0.93° and a high diffraction efficiency of 85% at 639 nm. In the experiment, a maximum AO Q-switched output power of 362 mW is achieved at a repetition rate of 30 kHz, under a maximum absorbed pump power of 3.60 W, corresponding to a slope efficiency of 15.2%. With transmittance of T = 3%, the shortest Q-switching pulse width of 54.7 ns is generated at a repetition rate of 1 kHz. Meanwhile, the beam quality factor $M^2$ of the above laser is measured, presenting the magnitude of 1.14 at both x and y directions. Our findings indicate that $\alpha$-BTM AO Q-switch could act as an excellent switching device at 639 nm which may help to explore potential applications in the visible field.

**Keywords:** $\alpha$-BaTeMo$_2$O$_9$ ($\alpha$-BTM); acousto-optic (AO) Q-switch; visible laser pulse





## 1. Introduction

Recently, lasers in the visible range have gained great attention in various fields, including optical data storage, underwater communications, scientific research, and material handling [1–4]. Thanks to the improvements in GaN laser diode (LD), all-solid-state visible lasers directly pumped by GaN LD have been developed rapidly owing to compactness and high efficiency. Similarly, large amounts of low-dimensional materials such as black phosphorus [5], Cr$_2$Si$_2$Te$_6$ [6], Nb$_2$GeTe$_4$ [7], GeAs$_2$ [8], Bi$_2$Se$_3$ [9], MoS$_2$ [10], SWCNT [11], and Mo$_2$C [12] have been utilized to generate Q-switching and mode-locking laser pulse in visible and near-infrared bands. For example, Cui et al. obtained a stable passively Q-switched (PQS) Pr:YLF laser by employing PtSe$_2$ thin films and achieved the shortest pulse width of 91.8 ns at 640 nm [13]. Shang et al. used TiS$_2$ to achieve a stable self-starting Q-switched fiber laser and obtained the shortest pulse duration and maximum output power of 1.45 μs and 3.93 mW, respectively [14].

Compared with PQS lasers, acousto-optic (AO) Q-switched lasers have the advantages of precise control, high efficiency, and stable operation [15,16]. As we know, Quartz, PbMoO$_4$, and TeO$_2$ crystals are the common AO mediums because of their excellent properties such as large figure of merit, high diffraction efficiency, and low acoustic attenuation value [17]. In 2020, Jin et al. successfully realized a visible laser by using a quartz AO switch with the shortest pulse duration of 81.1 ns and maximum pulse energy of 3.94 μJ [18]. Despite the progress, there are some drawbacks in the practical applications. For example, PbMoO$_4$ is a lead-containing compound that is harmful to the environment, and it was completed cleavage along c axis [19]. AO figure of merit is an important material property parameter to characterize the diffraction ability of AO media. In general, a large AO figure of merit implies a large diffraction efficiency. In addition, novel AO mediums with a wide

transmission range and high diffraction efficiency are demanded for applications in laser Q-switching devices and phase modulators.

Recently, a novel AO switch based on an $\alpha$-BaTeMo$_2$O$_9$ ($\alpha$-BTM) crystal has been designed and its excellent Q-switching laser performance was demonstrated in erbium- and ytterbium-doped fiber lasers [20]. As a novel AO medium, it was noteworthy that the diffraction efficiency and diffraction angle of the $\alpha$-BTM crystal at 1.06 $\mu$m were 82.1% and 1.432°, respectively, which was comparable to TeO$_2$ [21]. The $\alpha$-BTM crystal is a biaxial crystal which belongs to the orthorhombic system with index axes X, Y, Z parallel to the crystallography axes b, c, and a, respectively [22]. The sound velocities of $\alpha$-BTM crystal along the direction [100] was 4375 m/s [21], which was greater than that of TeO$_2$ crystal (4200 m/s) [23]. A large diffraction angle, high diffraction efficiency, and sound speed can help to compress the laser pulse width and improve the contrast ratio of the laser pulse signal. More importantly, it possesses wide light transmission ranging from 0.38 to 5.53 $\mu$m and a transmittance of about 80% [22]. The outstanding performance demonstrated by $\alpha$-BTM AO Q-switch has motivated us to explore Q-switching applications further in other bands.

In this paper, an $\alpha$-BTM AO Q-switched Pr:YLF solid-state laser is realized for the first time. Under a repetition rate of 1 kHz, the shortest Q-switching pulse width of 54.7 ns is generated.

## 2. Experimental Setups

Figure 1 illustrates the setup of an $\alpha$-BTM AO Q-switched Pr:YLF laser. The pump source was a fiber-coupled GaN LD with the central wavelength of 442 nm. The maximum output power was 3.6 W. The 0.5% Pr:YLF crystal with the dimensions of $3 \times 3 \times 8$ mm$^3$ was well polished on both faces. A copper block linked to a water-cooled chiller was utilized to increase the heat dissipation effect. The temperature of water-cooled chiller was maintained at 18 °C. A plane mirror was applied as input mirror with anti-reflection (AR) coated at 442 nm and high reflection (HR) coated around 640 nm. The curvature radius of the output couplers (OCs) was R = −100 mm and they had three different transmittances of 1%, 3%, and 5%, respectively.

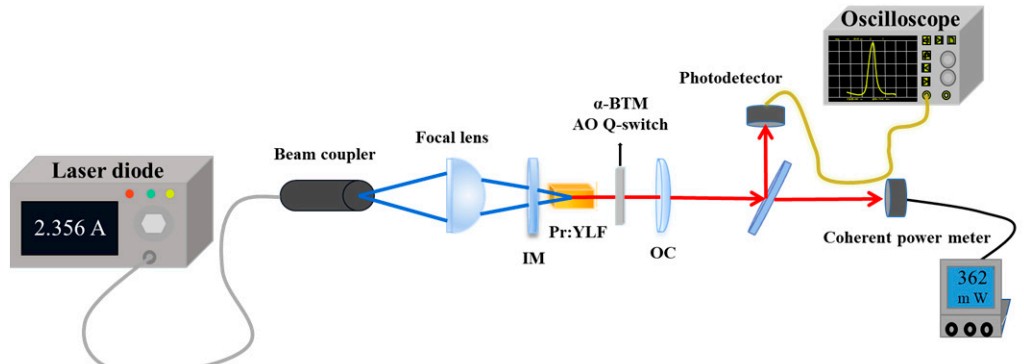

**Figure 1.** Experimental setups of the $\alpha$-BTM AO Q-switched Pr:YLF laser.

The average powers were measured with a Coherent power meter (Coherent: Field-mate Serial #0320A06). A visible CCD camera (Spiricon: Part NO. SP90421) was used to display the 2D distribution of the output beam. A digital oscilloscope (Rohde & Schwarz, RTO2012, 1 GHz bandwidth, 10 Gs/s sampling rates) equipped with a silicon detector (DET025A/M, 2 GHz bandwidth) was employed to detect the laser pulse signal.

Figure 2 displays a schematic diagram and a photo of the $\alpha$-BTM Q-switch. The source of the $\alpha$-BTM crystal is State Key Laboratory of Crystal Materials, Shandong University. The Q-switch mainly includes piezoelectric transducer, $\alpha$-BTM crystal, and impedance matching network. A Y36°-LiNbO$_3$ crystal with a thickness of 35 $\mu$m was used as a piezoelectric transducer to excite a pure longitudinal wave. The dimensions of the $\alpha$-BTM crystal are $14 \times 10 \times 5$ mm$^3$ and both sides are AR coated at 639 nm. The sound waves propagate along the crystallographic a-axis and light propagates along the crystallographic b-axis of

the $\alpha$-BTM crystal. The electrode layer and bonding layer are made of pure gold and pure tin, respectively, which provide a large electric load capacity and good heat transfer for AO medium.

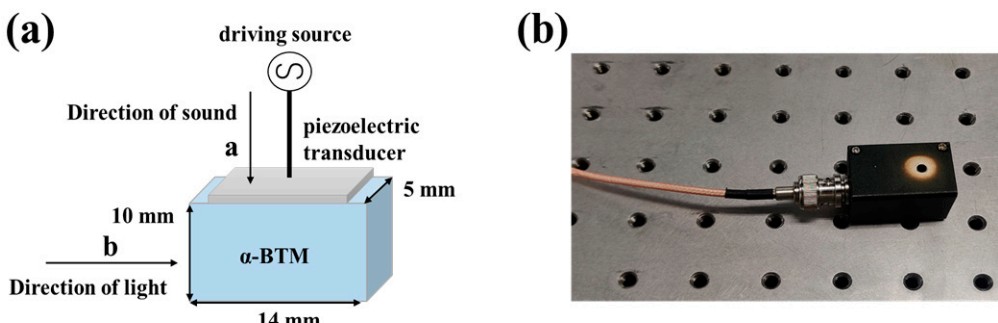

**Figure 2.** The schematic diagram (**a**) and photo (**b**) of the $\alpha$-BTM AO Q-switch.

### 3. Results and Discussions

First, the $\alpha$-BTM AO modulator was placed outside the resonator. The optical path was adjusted to make the incident light shine on the optical screen vertically. The diffraction angle and diffraction efficiency were measured using a home-made Pr:YLF continuous-wave (CW) laser. Figure 3 shows a schematic representation of an $\alpha$-BTM AO switch diffraction angle measurement at 639 nm. By measuring the distance of $L_1$ and $L_0$, the diffraction angle $\varphi$ can be calculated with the following formula

$$\varphi = arctan\left(\frac{L_1}{L_0}\right) \tag{1}$$

where $L_1$ is the distance between zero-level and first-level diffracted light and $L_0$ is the distance between zero-level and the $\alpha$-BTM AO switch. The diffraction efficiency $\eta$ was calculated using the following formula

$$\eta = \frac{P_1}{P_0} \times 100\% \tag{2}$$

where $P_1$ is the power of diffracted light of various levels and $P_0$ is the power of incident light. The diffraction angle of 0.93° and diffraction efficiency of 85% at 639 nm were calculated and compared with those of TeO$_2$ AO crystal.

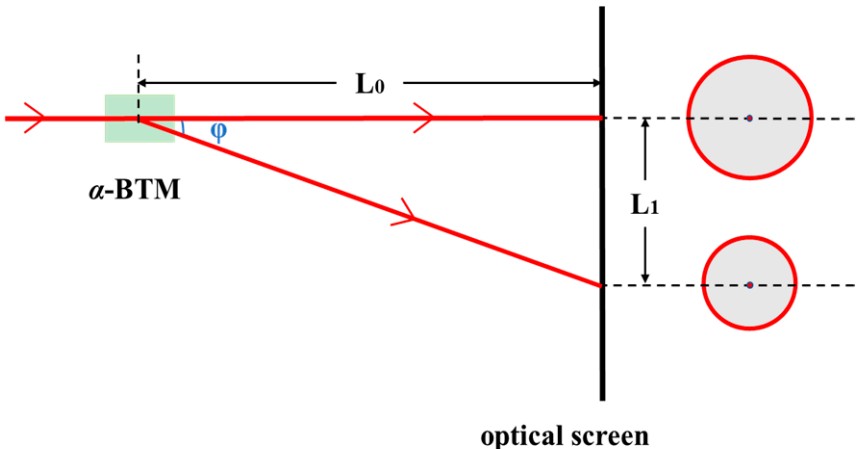

**Figure 3.** The schematic diagram for measuring the diffraction angle of the $\alpha$-BTM AO switch at 639 nm.

Without the Q-switch, the properties of the CW Pr:YLF crystal lasers were then studied by applying three OCs (T = 1%, 3%, and 5% are available). Figure 4a demonstrates the variety of CW output power with the absorbed pump power for various OCs. The absorption efficiency of the laser crystal in our experiment was found to be 84.4%. With an output transmittance of 5%, the maximum average output power of ~790 mW was achieved, corresponding to a slope efficiency of 29.2%. For the other OCs, the maximum output power dropped to 479 mW for T = 1% and 520 mW for T = 3%. The corresponding slope efficiencies were 17.2% and 19.0%, respectively. Compared with previous results, the maximum average output power of 790 mW is greater than the other LD end-pumped lasers [24–26], and the power of the CW laser is also stable enough to be measured with a power meter.

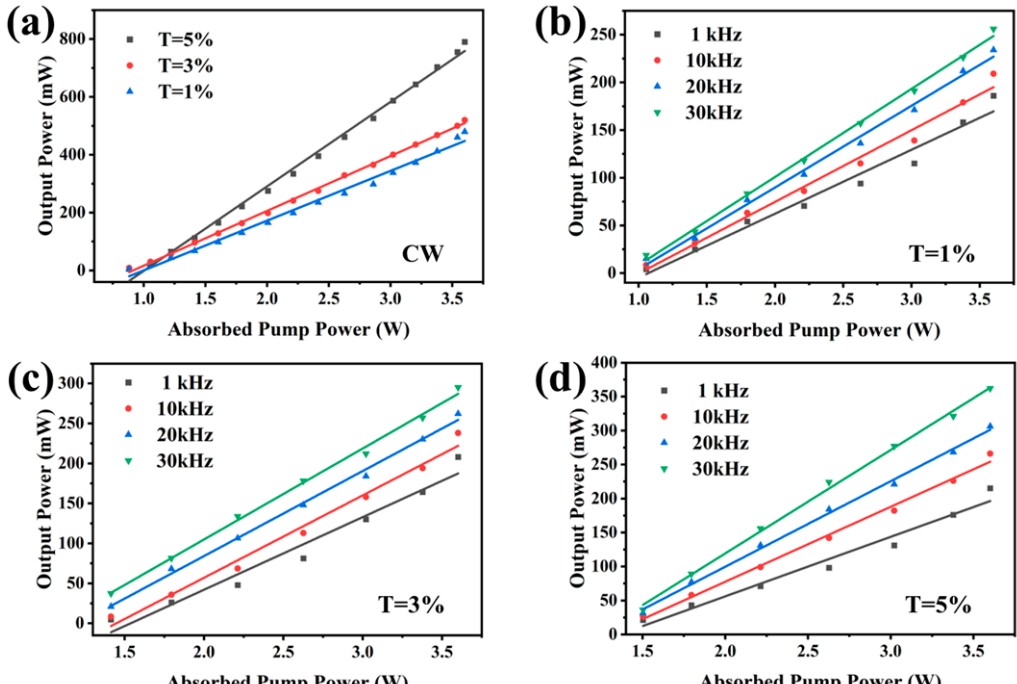

**Figure 4.** The dependences of CW and AO Q-switched output power on absorbed pump power, (**a**) CW; (**b–d**) Q-switched: b for T = 1%, c for T = 3% and d for T = 5%.

The laser characteristics of the $\alpha$-BTM AO Q-switched laser were also researched by employing these OCs. Figure 4b–d demonstrate the dependences of the average Q-switched output power on the absorbed pump power at different AO repetition rates. It is obvious that the output power is linearly and positively correlated with the absorbed pump power and $\alpha$-BTM Q-switching repetition rate. With the transmittance of T = 5%, the maximum Q-switched output power of 362 mW was obtained at a repetition rate of 30 kHz. The slope efficiency was 15.2%.

Figure 5a displays the shortest pulse widths versus the repetition rates. At the repetition rate of 1 kHz, the shortest pulse width of 54.7 ns was obtained under the absorbed pump power of 3.60 W. Figure 5b describes the variation of the shortest pulse widths with the absorbed pump power for different repetition rates under T = 3%. The relatively short laser pulse width can be attributed to the fast-propagation sound velocity, the large diffraction angle, and the high diffraction efficiency of the $\alpha$-BTM AO Q-switch at 639 nm. The laser pulse was further compressed if the pump power could be consistently increased. The peak power and pulse energy were calculated and plotted as a function of pump power, as presented in Figure 6. When an absorbed pump power was increased to 3.6 W, the highest pulse energy of 208 µJ and maximum peak power of 3.8 kW were obtained.

As illustrated in Figure 7a, the shortest single pulse profile and stable temporal pulse trains were obtained under the repetition frequency of 1 kHz. The pulse is burr-free and has good contrast, which originates from high diffraction efficiency and large diffraction angle of the $\alpha$-BTM AO Q-switch.

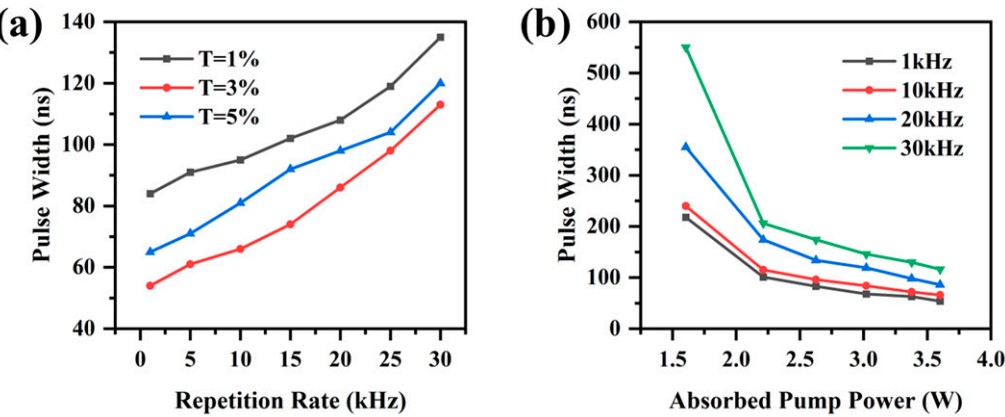

**Figure 5.** (**a**) Pulse widths versus repetition rates at the maximum absorbed pump power; (**b**) Variation of the shortest pulse widths with absorbed pump power at different repetition rates.

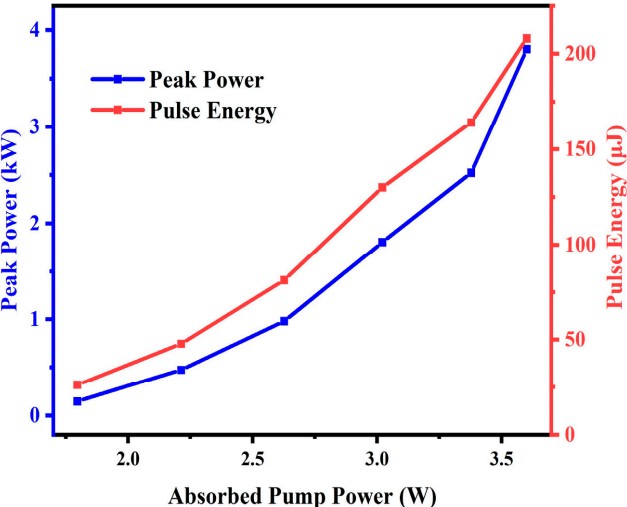

**Figure 6.** The dependences of pulse energy and pulse peak power on the absorbed pump power at T = 3%.

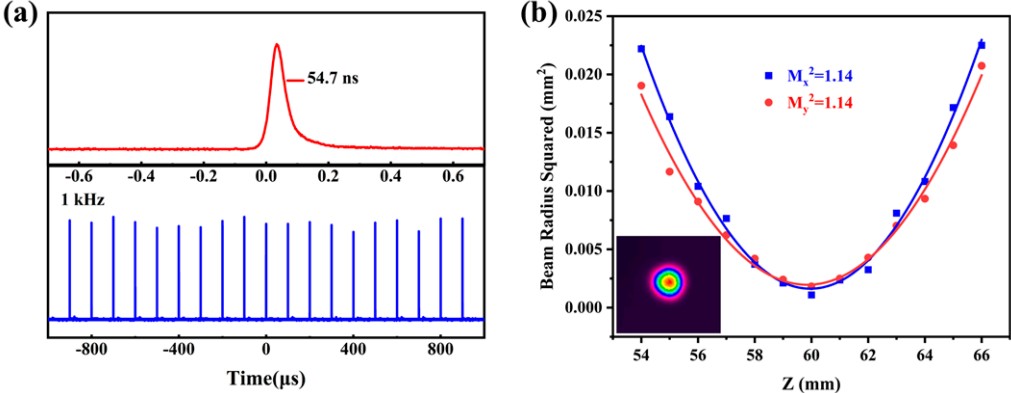

**Figure 7.** (**a**) The shortest single pulse profile and pulse trains; (**b**) Beam quality of the $\alpha$-BTM AO Q-switched Pr:YLF laser.

The knife-edge method was utilized to measure the beam quality factor $M_2$. Figure 7b shows the fitting line of the $M_2$ both in x-axis (vertical direction) and y-axis (horizontal direction). The fitting results show that the $\alpha$-BTM AO Q-switched Pr:YLF crystal laser operated on TEM$_{00}$ mode with $M_x^2$ of 1.14 and $M_y^2$ of 1.14, respectively. The 2D distribution of the output beam was inserted in Figure 7b, indicating an excellent TEM$_{00}$ mode Gaussian profile.

For comparison, Table 1 summarizes the Q-switching output characteristics of Pr-doped pulsed lasers at 639 nm. The shortest pulse duration of 54.7 ns was achieved with the $\alpha$-BTM AO Q-switch and the corresponding maximum peak power and highest pulse energy were 3.8 kW and 208 μJ, respectively. The excellent performance of the Q-switched laser is mainly due to the outstanding AO properties of the $\alpha$-BTM crystal at 639 nm, such as the fast sound propagation speed, high diffraction efficiency, and large diffraction angle. Further, the moderate specific heat and thermal conductivity can reduce the thermal effect on the $\alpha$-BTM crystal. In addition, the $\alpha$-BTM crystal was non-toxic and was easy to grow into high quality bulk single crystal. The results demonstrate that the $\alpha$-BTM crystal is a superior AO crystal for usage in the visible region and indicate that it has important optical applications in some specific wavelength ranges due to its wide bandwidth.

**Table 1.** The Q-switching output characteristics of Pr-doped pulsed lasers at 639 nm.

| Q-Switcher | Repetition Rate [kHz] | Pulse Width [ns] | Pulse Energy [μJ] | Peak Power [W] | Ref. |
|---|---|---|---|---|---|
| PtSe$_2$ | 297.6 | 91.8 | 0.057 | 0.622 | [13] |
| CdTe/CdS | 125.5 | 226 | 0.32 | 1.42 | [27] |
| EO | 0.1 | 137 | 260 | 1898 | [1] |
| AO (TiO$_2$ crystal) | 10 | $20 \times 10^3$ | 7.1 | – | [28] |
| AO (quartz crystal) | 10 | 81.1 | 3.94 | 48.5 | [18] |
| **AO ($\alpha$-BTM crystal)** | **1** | **54.7** | **208** | **3802** | **This work** |

## 4. Conclusions

In conclusion, an $\alpha$-BTM AO Q-switch was designed and utilized in a Pr:YLF laser. The diffraction angle and diffraction efficiency were measured by a home-made Pr:YLF continuous-wave laser, showing the diffraction angle of 0.93° and diffraction efficiency of 85%, respectively, which are competitive relative to those of the TeO$_2$ crystal. With the transmittance of T = 5%, the maximum Q-switched output power of 362 mW was obtained at the repetition rate of 30 kHz. At the repetition rate of 1 kHz, the shortest pulse width of 54.7 ns was obtained, corresponding to the maximum peak power of 3.8 kW and highest pulse energy of 208 μJ. These results demonstrate that the $\alpha$-BTM crystal is a potential AO Q-switch medium and has a great potential in broadband optoelectronics applications.

**Author Contributions:** Conceptualization, S.L., Z.G. and L.M.; methodology, S.L., X.T. and K.Z.; software, K.L., Y.J. and P.W.; validation, F.G., Z.G. and X.T; formal analysis, F.G. and P.W.; investigation, K.L. and L.M.; resources, S.L. and F.G.; data curation, P.W. and K.Z.; writing—original draft preparation, S.L. and K.Z.; writing—review and editing, S.L. and K.Z.; visualization, K.Z.; supervision, Z.G., L.M. and X.T.; project administration, F.G. and K.Z. All authors have read and agreed to the published version of the manuscript.

**Funding:** This work was financially supported by the National Natural Science Foundation of China (62175133), Natural Science Foundation of Shandong Province (ZR2020MF115), Shandong University of Science and Technology (2019TDJH103, skr21-3-049) and Project of Taishan Scholar (2021-175).

**Institutional Review Board Statement:** Not applicable.

**Informed Consent Statement:** Not applicable.

**Data Availability Statement:** Data are available upon request to the authors.

**Conflicts of Interest:** The authors declare no conflict of interest.

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
