# Peer review of "A Novel α-BaTeMo2O9 Acousto-Optic Switch for Generating Stable 639 nm Pulsed Laser"

_photonics, doi:10.3390/photonics10030334_

Round 1
Reviewer 2 Report
The authors of the current manuscript experimentally studied an acousto-optic (AO) Q-switch based on α-BaTeMo2O9 (α-BTM) crystal and further applied in generating laser pulse at 639 nm for the first time. The technical presentation of the results is comprehensive and easy to follow, but the following points require some additional clarifications before the paper can be accepted.
1. Did the authors measure the emission spectrum at the shortest pulse width of the Q operation?
2. In this work, the authors used an acoustic-optical Q-switch based on an α-BTM crystal to generate a 639 nm laser pulse. Could you describe more about the beam stability?
3. The authors obtained the maximum Q-switched output power in this experiment is 362mW, which I think that this result could be further improved.
4. When the absorbed pump power varied, how did the repetition frequency of the pulses change? What are the maximum and minimum repetition frequencies that the authors obtained in this experiment?
5. Whether the authors have investigated the ability of the acoustic-optical Q-switch based on α-BTM crystals to work in other wavelengths?
6. The introduction is too short, and the author is encouraged to introduce the knowledge of active and passive mode-locked or Q-switched lasers in the introduction. Encouraging the author to focus on recent work about passive mode-locked and Q-switched fiber lasers, such as, CST (10.1364/OME.446815), TMDs (10.1016/j.optlastec.2022.107988), Nb2GeTe4 (10.1021/acsnano.1c10241), GeAs2 (10.1002/adfm.202112252).
7. α-BTM crystal is a mature crystal, what are its advantages? The author should introduce more.
Reviewer 3 Report
Review: A novel α-BaTeMo2O9 acousto-optic switch for generating stable 639 nm pulsed laser
In the title of the manuscript, the stoichiometry of the compound is not well written.
The figure of merit mentioned in the introduction for AO materials should be defined, for large audience readers.
In the introduction, when describing the properties of the new AO medium, BTM, it is mentioned the high sound velocity along a crystallographic direction; so for better understanding of the reader; the description of the crystalline structure of the BTM crystal should be given before.
The source of the crystals used should be given. Also, their crystallographic orientation in their use(already given in the AO medium)
Please, justify briefly the choice of the laser medium.
Please, justify briefly the choice of the crystallographic orientation of the AO medium in the set up.
A more detailed description of the experimental measurement of the diffraction angle and efficiency should be given, in order to be able to be reproduced (beam spot dimensions, distance of the screen, error or accuracy of the precision; what the authors means as various levels?).
The results obtained in the CW lasing in the Pr:YLF crystals should be discussed in comparison with previous results in the literature.
When discussing the knife edge method, now x-axis and y axis are mentioned. What are these axes? Describe it briefly for the reader.
In table 1, what means EO? Is flint glass?
Please, add a discussion of the possible reasons for the high performance in the AO switch obtained with the BTM crystal.
Round 2
Reviewer 3 Report
if there are no restrictions of pages or words in the manuscript; all the detailed answers to the reviewers should be added in the manuscript.
